# Erectile Dysfunction Treatment Using Stem Cells: A Review

**DOI:** 10.3390/medicines8010002

**Published:** 2021-01-06

**Authors:** Vassilis Protogerou, Dimosthenis Chrysikos, Vasileios Karampelias, Ypatios Spanidis, Sara El Bisari, Theodoros Troupis

**Affiliations:** 1Department of Anatomy, School of Medicine, National and Kapodistrian University of Athens, Mikras Asias 21 st, 124 62 Athens, Greece; dixrys@yahoo.gr (D.C.); sara.bisari@hplus.ae (S.E.B.); ttroupis@med.uoa.gr (T.T.); 23rd Urological Department, School of Medicine, National and Kapodistrian University of Athens, Attikon University Hospital, Rimini 1, 124 62 Athens, Greece; 3Department of Surgery, School of Medicine, University of Patras, 265 04 Patras, Greece; vasiliskarampelias@hotmail.com (V.K.); akis.span@yahoo.gr (Y.S.)

**Keywords:** erectile dysfunction, stem cells, platelet rich plasma (PRP), platelet lysate plasma (PLP), adipose tissue-derived stem cell (ADSC), bone marrow-derived stem cell (BMSC), novel treatments

## Abstract

Erectile dysfunction (ED) is a disorder that affects the quality of life and the sexual relations of more than half of the male population aged over 40 years. The prediction regarding the incidence of ED is devastating as it is expected that this disorder will affect more than 300 million men in the next five years. Several studies have suggested the use of stem cells for the treatment of ED and showed that this type of treatment is promising in terms of damaged tissue repair as well as of clinical efficacy; however, there are several gaps in the knowledge and evidence is lacking. In order to highlight a few of them in this review, we performed a research of the literature focusing on currently available clinical studies regarding the clinical efficacy of stem cell administration for the treatment of ED. We reviewed the methods of administration, the cell types used in the performed clinical trials and the safety and efficiency of such procedures. We conclude that there are rapidly expanding and promising results from the reported clinical studies indicating that stem cells could indeed be a potential treatment for patients with ED although more studies are necessary.

## 1. Introduction

Erectile dysfunction (ED) is a common condition that refers to the inability of a male individual to attain and maintain sufficient penile erection for sexual intercourse [1]. It is classified as organic, psychogenic or neurogenic [2]. Over the past years, ED has raised increased concern and has been recognized as a public health problem [3]. It mainly affects men aged > 40 years and has a significant impact on their sexual life [4]. Interestingly, the Massachusetts Male Aging Study indicated that approximately 52% of 1290 men aged 40–70 years suffered from ED [5]. Future projections show a prevalence of 322 million men with ED by 2025, indicating an increase of more than 100% compared with the corresponding rates in 1994 (approximately 154 million men with ED) [5]. There are several disorders that are linked to ED such as diabetes mellitus, metabolic syndrome, cardiovascular diseases, Parkinson’s disease and pelvic nerve injury [6]. The majority of these disorders are associated with endothelial dysfunction, which is linked to the severity of ED as the corpora cavernosa vascular homeostasis is mainly regulated by the vascular endothelium [7].

Other factors that play a significant role in the development of ED are the damage of the nerves that are responsible for the erection, side effects due to drug medication and alterations in hormone levels such as testosterone [8].

Until recently, the treatment of ED has mainly been based on the transient enhancement of penile erection but without a permanent reversal of endothelial dysfunction or restoration of the disturbed penile tissue homeostasis. Current treatment options include phosphodiesterase type-5 inhibitors (PDE5is) such as vardenafil, avanafil, tadalafil and sildenafil, which are the most widely known and used medications for the treatment of ED. Their mechanism of action is mediated by the Nitric Oxide (NO) cyclic guanosine monophosphate (NO-cGMP) pathway activation. In brief, penile erection requires arterial smooth muscle cells (SMC) relaxation to facilitate blood flow in the penile tissue (i.e., a network of arteries, capillaries and lacunae surrounded and supported by elastic tissue, collagen and muscle fibers) in order to produce penile engorgement and rigidity of the penis. The main metabolite inducing SMC relaxation is NO provided by the nonadrenergic noncholinergic postganglionic parasympathetic neurons and the endothelium. NO facilitates the transformation of inactive GTP to active cGMP, which eventually results in Ca++ decrease within the cytosolium, thus enhancing SMC relaxation in the corpus cavernosum causing vasodilation, blood inflow and erection. PDE5 enzymes metabolize cGMP to an inactive form so PDE5is inhibit the degradation of cGMP and the final event is a more pronounced decrease in Ca++, which facilitates erection [8,9]. Of note, the last ten years of reports have expanded our knowledge by indicating that PDE5is also stimulate bone marrow endothelial progenitor cell function and inhibit the apoptosis of smooth muscle cells [10,11,12,13,14].

Unfortunately, the treatment failure levels with PDE5is are unacceptably high. Among patients using PDE5is, approximately 20% of those who had ED without other comorbidities and 40% of those with diabetes mellitus or underwent a prostatectomy remained unresponsive to these treatment procedures [15]. Therefore, additional treatment options for the management of ED are needed and include the use of intracorporal injections, vacuum erection devices and penile prosthesis implantation. However, their application is also limited due to the high cost, intolerance to side effects, pain and unsatisfactory results [8]. Consequently, the need to develop efficient curative treatments for patients with ED turned the scientific interest toward the study of stem cell therapy [6].

In this systematic review, we aim to shed light into the gaps in knowledge and evidence regarding the potential use of stem cells for ED treatment and provide an overview of the currently available knowledge and concerns.

## 2. Stem Cell Therapy

Stem cells are undifferentiated or partially differentiated cells, which are characterized by their ability to self-renew and differentiate into more specialized cell types [16]. Generally, stem cells are classified as totipotent (e.g., zygote), pluripotent (e.g., embryonic stem cells (ESCs)), multipotent (e.g., hematopoietic and mesenchymal stem cells (MSCs)) and unipotent according to the number of cell lines in which they could be differentiated [17]. When a stem cell is divided, the daughter cells can either remain stem cells or differentiate into a specialized cell type, e.g., muscle or nerve tissue cells [18]. Their ability for division, differentiation and tissue regeneration is affected by their environment, which supports the stem cells and interacts with them, affecting their transformation to dedicated cell types and promoting their self-renewal [19]. The regenerative effects of stem cells are attained by the secretion of numerous growth factors (paracrine action) and/or their migration to the injury site along with cell contact and cellular differentiation [20]. In these sites they could regenerate the damaged tissues according to the stimuli or the received signals [18].

Stem cells have been used for the treatment of many diseases since the 1990s and have shown promising potential, thus allowing scientists to consider their use as a plausible candidate approach [21,22]. Indeed, their importance is reflected by their wide application as their anti-inflammatory, restorative and immunomodulatory properties are linked with the treatment of cardiovascular [23], neurological [24], autoimmune [25] and hematologic diseases [26,27], Parkinson’s disease [28], strokes [29] and spinal cord injuries [30]. It is possible that diseases or pathological conditions of the urinary system would also benefit from the use of stem cells.

In recent years, stem cell therapy has been proposed for the treatment of ED as stem cells can differentiate to endothelial, neuronal or smooth muscle cells and therefore restore possible structural damage in the penile tissue. In vitro stem cell differentiation in these cell lines has been proven while preclinical studies showed improvement in ED following stem cell therapy in several animal models [31,32,33]. However, the exact mechanism has not yet been proven and also includes the paracrine action of stem cells as another possible mechanism in ED shown in animal models [32].

Stem cells that have been used in ED preclinical studies are mainly adipose tissue-derived stem cells (ADSCs) [31], bone marrow-derived stem cells (BMSCs) [33], muscle-derived (MDSCs) [34] and ESCs [16].

Clinical studies with the use of stem cells have also been published with encouraging results [21,35,36,37,38,39,40,41,42,43]. Questions regarding the clinical use of stem cells include the type of stem cells, the method of preparation, the optimal number of stem cells and the method of administration.

## 3. Stem Cell Delivery

Different ways of stem cell administration have been suggested and examined. Specifically, reported preclinical works have examined the effect of intraperitoneal and intravenous injection and suggested that the latter was more efficacious in improving erectile function [44,45,46]. Periprostatic implantation has also been performed in other works and the results depicted an equal efficacy with an intracavenosal injection [47,48]. In general, a direct injection into the organ of concern has been performed and suggested by many studies [21,35,38] despite the fact that lower than 1% of the injected stem cells might remain in the target tissue and could finally dissipate at only few days after injection [49]. Although only a limited number of stem cells remained in the target tissues after injection, they created a significant effect by triggering endogenous mechanisms of regeneration and promoted the propagation and differentiation of progenitor cells, thus improving the recovery of the target tissue [46,49,50,51].

Regarding ED treatment, in all of the clinical studies so far stem cell delivery has been studied by direct injection in the target tissue of the penis or an intracavernosal injection.

## 4. Clinical Studies: Type of Stem Cells, Preparation and Efficacy in ED

In the clinical studies published so far, the stem cells used have been BMSC, ADSC, umbilical cord stem cells and placenta-derived stem cells [21,35,36,37,38,39,40,41,42,43]. In terms of preparations, there have been two basic techniques to prepare stem cells. One is the isolation of the stem cells from the tissue obtained from the donor and then used directly or expanded in culture to obtain greater numbers of stem cells. The other is in the form of Stromal Vascular Fraction (SVF) and applies to ADSC. SVF is a product of the adipose tissue obtained by the donor and it is created in a process that includes centrifugation of the fatty tissue harvested. SVF is a product of the centrifugation and contains stem cells but also endothelial precursor cells, growth factors and immune modulatory cells [52]. Although by expanding the stem cells you can obtain a greater number compared with stem cells in SVF, the additional cells and factors included in the SVF seem to collaborate with the stem cells and the final clinical result might be even better although no final conclusion has been made [53]. An overview of clinical studies is presented in Table 1.

The first clinical study was published in 2010 by a Korean team. Bahk et al. in this pioneered study innovated because the stem cells used were provided by a company and not harvested from patients. Umbilical cord stem cells (1.5 × 10^7^) were injected in seven men aged 57–83 years with diabetes-related ED [35]. The results revealed that the majority of their participants regained their morning erections within one month and maintained this for more than six months. Moreover, their blood glucose levels decreased after two weeks, highlighting that human umbilical cord blood stem cell therapy provided positive outcomes in both ED and diabetes conditions.

In 2013, the first BMSC administration in the treatment of ED patients was reported as a case report by Ichim et al. [37]. A 35 year old patient with a past medical history of smoking and hypercholesterolemia was suffering from ED and he was unresponsive to PDE5 inhibitors. He was treated with BMSC without specifying the number of stem cells used. There was a return in spontaneous erections but he still required medication (PDE5is) to sustain them until orgasm. The beneficial effect was present 18 months post-treatment.

Another significant study from 2016 evaluated for the first time the use of various stem cell numbers per injection. Yiou et al. in phase I/pilot study reported the effects of a BMSC injection in patients with vasculogenic ED who had undergone a radical prostatectomy [38]. Specifically, they divided their participants in four groups and administered escalating doses of stem cells (2 × 10^7^, 2 × 10^8^, 1 × 9^9^ or 2 × 10^9^ stem cells). Their results revealed that a significant improvement without serious side effects was observed in the patients who received the highest dose of stem cells at six months post-treatment, which was associated with improvements in their peak systolic velocity measured by penile triplex. The same team published a phase II study in 2017, stating that the optimal dose was found to be the 1 × 9^9^ dose (43).

Adipose-derived stem cells (ADSCs) are increasingly used over BMSC currently in regenerative medicine. ADSCs are a distinct population of MSCs, which reside in the perivascular niche of the adipose tissue [54]. The broad use of these cells could be attributed to their easy isolation by specific devices, their abundance in the tissues, their similar properties to BMSC and their proven therapeutic efficacy [15]. The isolation of such cells demands firstly the separation of the SVF from the adipose tissue harvested, as already mentioned. The SVF can then be used either alone or alternatively stem cells can be separated from it and expanded under culture. In a small study in 2015, Garber et al. administered 1.5 × 10^7^ adipose stem cells in six diabetic patients [36]. Stem cells were obtained after culture and not in SVF form. A return of morning erections was noted in four patients in the first month whereas by the twelfth month, four out of six patients were able to have sex with the use of PDE5 inhibitors. These results were in line with those by a Danish team published in 2018 [21]. In this study, SVF was used. Specifically, Haahr et al. showed that an intracavernosal injection of 8.4–37.2 million freshly isolated autologous ADRCs was a safe and effective procedure in patients with ED who had prostate cancer and had undergone a radical prostatectomy (RP). Moreover, no serious adverse effects were observed during a 12-month follow up and most of the participants recovered erectile function within six months and this effect persisted for the 12-month follow up period. In a recent novel work by our team [39,42], a combined technique of delivering stem cells was introduced. We administrated adipose-derived MSCs obtained after culture, resuspended in platelet lysate plasma (PLP). In that way, the high numbers of stem cells obtained by culture were mixed with the growth factors of PLP in an effort to mimic the beneficial effects of the growth factors of SVF and at the same time create larger numbers of stem cells. The results were promising as the treatment improved the erectile function of patients after one, three and six months of follow up without presenting any negative side effects. This is so far the only study that has combined stem cells with PLP.

The efficacy of a placental matrix-derived stem cell injection in patients with ED was also reported in a 2016 study by Levy et al. [41]. Specifically, they injected 1.5 mL of placental matrix-derived stem cells into the base of the corpora cavernosum. The results were satisfactory as the patients improved peak systolic velocity measured with penile triplex performed six months after the injection. An improvement in erections appeared six weeks post-treatment and erections were attained for six months after PDE5is administration.

The re-administration of stem cells was studied in 2018 [40]. Al Demour et al. confirmed the efficacy, safety and tolerability of repeated intracavenous autologous bone marrow-MSC injections in four patients with ED and diabetes. Two treatments of 30 × 10^6^ cells each were offered within 30 days. An improvement in the IIEF-5 score was noted without any side effects from the two treatments.

Overall, the clinical studies published so far provide encouraging results. Improvement of sexual function is reported with no side effects. Unfortunately, although pioneering, all of these are small studies with a short follow up period, various etiologies of ED and without a control group.

## 5. Side Effects

Overall, there were no side effects reported. Bahk et al. [35] reported no side effects. Ichim et al. [37] reported no immediate injection-associated side effects and Garber [39] also reported no adverse effects. Yiou et al. [38] reported only side effects from the aspiration site of the bone marrow mononuclear cells. Specifically, they reported mild postoperative pain at the bone marrow aspiration site that was not present on the first month after at the follow up visit. He also stated that there were no cases of priapism. Microbial growth was noted in the samples taken from three patients but without any clinical side effects. A decrease in hemoglobulin was noted due to the aspiration of the cells but no blood transfusion was needed. There were no increases in PSA values nor changes in the digital rectal examination in the patients post-treatment. Haahr et al. [21] reported only mild effects on the injection site (transient redness and swelling, scrotal and penile hematomas) or from the liposuction site; all of them recovered spontaneously. Levy et al. [41] reported also mild irritation on the injection site that resolved in 48 h. Yiou et al. [43] in their follow up study with the long term results once more reported no side effects and no prostate cancer recurrences at five years of follow up. Al Demour et al. [40] reported only minor side effects from the aspiration and injection site with no side effects in two years of follow up regarding nervous, cardiovascular, respiratory and gastro-intestinal systems. Finally, Protogerou et al. [39,42] reported only minor side effects from the injection site.

## 6. Future Implementation and Perspectives

A double-blind randomized controlled study is urgently needed for the investigation of the clinical effect of stem cell treatment in ED. Unfortunately, due to its invasive nature, it is not easy to organize. Nevertheless, the results are encouraging so far but questions need to be answered such as: which is the ideal stem cell for the treatment of ED? What is the optimal dose? Should we use expanded cultured cells or SVF from the adipose tissue? Or should we combine expanded cells with PLP to get the best of the two treatments? Most importantly, should we use a patient’s own stem cells or instead should we use cells from a younger, healthier donor?

## 7. Conclusions

Currently, several preclinical and clinical studies have reported preliminary data regarding the safety and potential efficacy of stem cell treatment in ED. Nevertheless, future randomized clinical trials examining their use in an adequate number of humans over a long period of time using different templates of doses and numbers of injections are needed to be performed to obtain sufficient knowledge regarding the ideal therapeutic strategy and elucidate any potential adverse effects.

## Figures and Tables

**Table 1 medicines-08-00002-t001:** Overview of clinical studies.

Year Of Publication (Reference Number)	Stem Cell Used	Additional Treatment	Latent Period until Erectile Stimulation Was Detected	Period during Which Erectile Stimulation Was Maintained
2010 (35)	Umbilical cord stem cells	No	1 month	11 months
2013 (37)	BM mononuclear cells	No	3 weeks	18 months
2015 (34)	ADSC	No	1 month	12 months
2016 (38)	BM mononuclear cells	No	3 months	6 months
2016 (41)	Placental matrix-derived stem cells	No	6 weeks	6 months
2017 (43)	BM mononuclear cells	No	3 months	6 months
2018 (21)	ADSC	No	6 months	12 months
2018 (40)	BMSC	Re-administration of 2nd injection	1 month	12 months
2019 (39)	ADSC	PLP	1 month	3 months
2020 (42)	ADSC	PLP	1 month	6 months

BM: Bone marrow, ADSC: Adipose Derived Stem Cells, BMSC: Bone Marrow Stem Cells, PLP: Platelet Lysate Plasma.

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
