# Peer review of "Erectile Dysfunction Treatment Using Stem Cells: A Review"

_medicines, 2021, doi:10.3390/medicines8010002_

Round 1
Reviewer 1 Report
Authors answered to all my concerns.
Author Response
Dear Reviewer 1
Thank you for your positive comments.
Your Sincerely
Dr V. Protogerou
Reviewer 2 Report
The authors have attempted to respond to our comments and suggestions as detailed below. We have reviewed each response individually. The overarching concerns and criticisms, however, remain unaddressed. There continues to be persistent issues with grammar and syntax. If the authors want to publish in an English speaking journal, they must employ someone with appropriate English writing capabilities to assist. In addition, our belief remains that larger studies with adequate controls are needed to be performed before a review like this can be published.
Regarding responses to reviewer comments, the authors need to be more specific about where they made their changes in the article. They should employ tracked changes to help a reviewer identify where edits have been made. The brief author comments and highlighting the text in yellow are not adequate.
1) Page 1, Lines 47-48: This should read, “Until today the treatment of ED is mainly based on creating an erection rather than treating the affected penile tissue.”
Author comment: We changed lines 47-48 according to your instructions.
Reviewer comment: “Untill” is still spelled incorrectly.
2) Page 1, Line 51: The authors need to improve on this explanation of the mechanism of erectile dysfunction (ED). It does not read well and is confusing to the reader.
Author comment: We add more details in erectile mechanism in line 51 according to your comments.
Reviewer comment: This reads better; however, there is still poor syntax (e.g., “smooth muscles that surround (the) artery.”)
3) Page 1, Line 56: Remove “the” before “bone marrow.”
Author comment: We changed line 56.
Reviewer comment: Accepted
4) Page 1, Lines 84-86: The authors write, “In recent years, stems cell therapy is proposed for the treatment of ED by replacement of the lost or damaged cells, threatened host cells protection, provision of trophic factor, or gene delivery.” The authors must elaborate more here, as these points are fundamental to understanding the proposed mechanisms. What are the lost or damaged cells suggested? Are they smooth muscle, capillaries/blood vessels, or non-cholinergic, non-adrenergic (NCNA) neurons? How do stem cells actually regenerate these tissues in the penis? Of course, a lot is unknown regarding the mechanisms, but a postulation or mechanistic proposal is needed.
Author comment: We changed the lines 84-86 to be more explanative as you suggested.
Reviewer comment: Again, there is poor syntax.
5) Page 1, Line 86: I do not understand what “threatened host cells protection” means. Please elaborate.
Author comment: We rearranged line 86.
Reviewer comment: Accepted
6) More information is needed on the proposed mechanisms of the provision of trophic factors and gene delivery.
Author comment: We rearranged this paragraph.
Reviewer comment: Accepted
7) Section 4: Most of the studies cited are in very small study populations (<10 subjects).
Author comment: Indeed, the sample size of the studies is small but these are the only studies that exist.
Reviewer comment: As mentioned previously in my initial review, until there are better stem cell papers addressing the problem of ED with adequate controls, I do not think it worthwhile to publish a literature review.
Author Response
Dear Reviewer 1,
After your comments we made the following changes:
1) Page 1, Lines 46: We changed “Untill” to Until
2) Page 1, Line 51: We completely rearranged the paragraph
4) We did extensive changes in the manuscript according to your instructions.
7) Section 4. We understand your comment regarding the small sample size of the studies. Nevertheless, these are the only existing studies in this field, and we disagree that we should not review them. Progress in science comes in small steps and maybe without pioneering small studies, bigger one might not be possible to be organized. One of these studies was presented at the 2017 Congress of European Association of Urology and drew a lot of scientific attention due to its innovation. Moreover, we state in our Conclusion section that more studies are needed with more patients.
Yours Sincerely
Dr V. Protogerou
Reviewer 3 Report
Excellent study with no major weaknesses. will be widely quoted, leading study in the field.
Author Response
Dear Reviwer 3
Thank you for your positive comments
Yours Sincerely
Dr V. Protogerou
This manuscript is a resubmission of an earlier submission. The following is a list of the peer review reports and author responses from that submission.
Round 1
Reviewer 1 Report
The authors reviewed the literature focused on the currently available clinical studies regarding the efficacy of stem cell administration for the treatment of erectile dysfunction. However, there have been many similar articles published recently and no original views presented by the authors.
I have to say there are no merits to publish this review in a scientific journal.
Reviewer 2 Report
In this review, the Authors explored the role of stem cell therapy in erectile dysfunction. Despite the promising results of the current trials in literature, they concluded that further are needed, with a larger population and longer follow-up.
I appreciate this work, and I have only minor concerns:
- lines 45-46: this is the aim of the study, I suggest to move at the end of the Introduction;
- Please, could Authors specify the study design (e.g. a narrative review, systematic review, etc.);
- I suggest the Authors include a specific section on the side effects of this treatment.
Author Response
- We put the lines 45-46 at the end of Introduction section.
- Regarding the very few papers on the field it is a systematic review.
- We added a Side Effects section according to your instructions
Reviewer 3 Report
In this article, the authors have attempted to review the current literature with respect to the use of stem cells in erectile dysfunction treatment. There are also some fundamental additions that are needed prior to suggesting publication, including more detail on the proposed mechanisms of stem cell regeneration in particular with respect to tissues of the penis.
1) Page 1, Lines 47-48 – This should read, “Until today the treatment of ED is mainly based on creating an erection rather than treating the affected penile tissue.”
2) Page 1, Line 51 - The authors need to improve on this explanation of the mechanism of erectile dysfunction (ED). It does not read well and is confusing to the reader.
3) Page 1, Line 56 - Remove “the” before “bone marrow.”
4) Page 1, Lines 84-86 – The authors write, “In recent years, 84 stems cell therapy is proposed for the treatment of ED by replacement of the lost or damaged cells, 85 threatened host cells protection, provision of trophic factor, or gene delivery.” The authors must elaborate more here, as these points are fundamental to understanding the proposed mechanisms. What are the lost or damaged cells suggested? Are they smooth muscle, capillaries/blood vessels, or non-cholinergic, non-adrenergic (NCNA) neurons? How do stem cells actually regenerate these tissues in the penis? Of course, a lot is unknown regarding the mechanisms, but a postulation or mechanistic proposal is needed.
5) Page 1, Line 86 - I do not understand what “threatened host cells protection” means. Please elaborate.
6) More information is needed on the proposed mechanisms of the provision of trophic factors and gene delivery.
7) Section 4: Most of the studies cited are in very small study populations (<10 subjects).
Author Response
- We changed lines 47-48 according to your instructions.
- We add more details in erectile mechanism in line 51 according to your comments.
- We changed line 56
- We changed the lines 84-86 to be more explanative as you suggested.
- We rearranged line 86.
- We rearranged this paragraph
- Indeed, the sample size of the studies is small but these are the only studies that exist.
Reviewer 4 Report
well written paper.
The discussion in the stem cell therapy section should have a more elaborate description of how the stem cells are thought to work. It says
"stem cell therapy is proposed for the treatment of ED by replacement of the lost or damaged cells, threatened host cells protection, provision of trophic factor or gene delivery." Several references are cited. The authors should write at least one paragraph explaining how MSCs work. what I quoted here does not really convey any useful information
Author Response
We add a paragraph with the mechanism of action of stem cells in ED according to your instructions